# Efficient Removal of Hexavalent Chromium (Cr(VI)) from Wastewater Using Amide-Modified Biochar

**DOI:** 10.3390/molecules28135146

**Published:** 2023-06-30

**Authors:** Ashraf Ali, Sarah Alharthi, Nora Hamad Al-Shaalan, Alia Naz, Hua-Jun Shawn Fan

**Affiliations:** 1Department of Chemistry, Faculty of Physical & Applied Sciences, The University of Haripur, Haripur 22620, Pakistan; 2Center of Advanced Research in Science and Technology, Taif University, P.O. Box 11099, Taif 21944, Saudi Arabia; 3Department of Chemistry, College of Science, Taif University, P.O. Box 11099, Taif 21944, Saudi Arabia; 4Department of Chemistry, College of Science, Princess Nourah bint Abdulrahman University, P.O. Box 84428, Riyadh 11671, Saudi Arabia; 5Department of Environmental Science, Faculty of Physical & Applied Sciences, The University of Haripur, Haripur 22620, Pakistan; aliaawkum@gmail.com; 6College of Chemical Engineering, Sichuan University of Science and Engineering, Zigong 643099, China

**Keywords:** biochar, amide-modification, wastewater treatment, kinetics, adsorption isotherm

## Abstract

The utilization of biochar, derived from agricultural waste, has garnered attention as a valuable material for enhancing soil properties and serving as a substitute adsorbent for the elimination of hazardous heavy metals and organic contaminants from wastewater. In the present investigation, amide-modified biochar was synthesized via low-temperature pyrolysis of rice husk and was harnessed for the removal of Cr(VI) from wastewater. The resultant biochar was treated with 1-[3-(trimethoxysilyl) propyl] urea to incorporate an amide group. The amide-modified biochar was characterized by employing Fourier transform infrared (FTIR) spectroscopy, scanning electron microscopy (SEM), and X-ray diffraction (XRD) techniques. During batch experiments, the effect of various parameters, such as adsorbent dosage, metal concentration, time duration, and pH, on Cr(VI) removal was investigated. The optimal conditions for achieving maximum adsorption of Cr(VI) were observed at a pH 2, an adsorbent time of 60 min, an adsorbent dosage of 2 g/L, and a metal concentration of 100 mg/L. The percent removal efficiency of 97% was recorded for the removal of Cr(VI) under optimal conditions using amide-modified biochar. Freundlich, Langmuir, and Temkin isotherm models were utilized to calculate the adsorption data and determine the optimal fitting model. It was found that the adsorption data fitted well with the Langmuir isotherm model. A kinetics study revealed that the Cr(VI) adsorption onto ABC followed a pseudo-second-order kinetic model. The findings of this study indicate that amide-functionalized biochar has the potential to serve as an economically viable substitute adsorbent for the efficient removal of Cr(VI) from wastewater.

## 1. Introduction

With the increasing human population and industrialization of heavy metals (HM), pollution poses a serious threat to human and aquatic environments. Numerous industries, including fertilizer, tanneries, battery manufacturers, paper production plants, and pesticide manufacturing operations, release hazardous materials into the environment. Owing to the non-biodegradable nature of heavy metals, they remain in soil and water for a longer time, accumulate in the food chain, and thus enter human bodies via metal-contaminated food, drinking water, and air [1]. Some HMs, such as chromium (VI), mercury (II), and lead (II), are toxic even in trace amounts, can damage the liver, central nervous system, lungs, kidneys, and skin, and can cause cancer [2,3]. Chromium is used in electroplating, leather, textile, and tanning industries, and these industries discharge chromium-rich effluents into the nearby waterbodies, thus contaminating the environment [4,5,6,7].

In aqueous media, chromium exists in two forms: Cr(III) and Cr(VI). The oxidation number of chromium depends on the pH value of the aqueous medium, and changing the pH can change chromium from one form to another [8,9]. The toxicity of chromium for humans is dependent on the exposure duration and dose, and it has been found that hexavalent chromium is more toxic than trivalent. Long-term exposure to chromium can damage the skin, eyes, blood, immune system, and respiratory system [10,11]. Chromium also damages DNA, creates oxidative stress, and can cause the development of tumors in the human body [12]. As per the guidelines established by the World Health Organization (WHO), the upper permissible threshold of chromium concentration in surface water and potable water is set at 100 µg L^–1^100 µgL^–1^ and 50 µgL^–1^, respectively. On the other hand, 500–270,000 µgL^–1^ Cr has been reported in industrial wastewater [13,14,15,16]. Owing to the high toxicity of chromium, it is necessary to develop an efficient method for its removal from wastewater before discharging it to the aquatic environment and dumping sites [17].

Various methods have been reported for the removal of Cr from wastewater, such as ion exchange, electrocoagulation, membrane filtration, precipitation, reverse osmosis, chemical precipitation, ultra-filtration, electrodialysis, electrochemical separation, coagulation, and adsorption [18,19,20,21,22,23,24,25,26]. Some of these methods, such as electrodialysis, ultra-filtration, chemical precipitation, coagulation, and electrochemical separation, have a high cost, low efficiency, consumption of energy, require a large number of chemicals, and produce secondary waste or sludge; therefore, these methods are not useful for industrial wastewater treatment. On the other hand, adsorption is preferred over other separation methods owing to its lower cost, simple design, high efficiency, less consumption of energy, and the production of fewer secondary wastes [21,27,28,29,30].

Various adsorbents have been employed for the elimination of heavy metals (HM) from wastewater, including activated carbon, ion exchangers, composite materials, nanoparticles, natural and synthetic polymers, as well as agricultural wastes like rice husk, wheat straw, sawdust, and fruit peels [31,32,33]. However, commercial adsorbents are expensive, and their high cost limits their use for wastewater treatment. In addition to the high cost, the regeneration of these adsorbents involves complex procedures which degrade their functionality, and thus, these adsorbents cannot be used several times for adsorption. Industries are currently facing significant demand for the development of an adsorbent that is both efficient and economically viable, while also being environmentally sustainable to facilitate the removal of chromium and other heavy metals from wastewater.

The utilization of biochar, which is comprised of diverse functional groups, such as hydroxy, carbonyl, carboxylic, and epoxy groups, has the potential to serve as a substitute adsorbent for expansive adsorbents like activated carbon for the purpose of eliminating toxic heavy metals (HM) from wastewater [34,35]. The characteristic features of porosity and functional groups render biochar a valuable substrate for the purpose of eliminating hazardous materials and various other types of pollutants [36,37]. The presence of diverse functional groups within biochar operates to facilitate the adsorption of metal ions and other pollutants through either electrostatic attraction or complexation. These properties have made biochar an ideal candidate and cheap alternative to the commonly used expansive adsorbents, such as ion exchange resin, activated carbon, and carbon nanotubes, for wastewater treatment [38,39]. The adsorption capacity of biochar may be enhanced through chemical functionalization with varying ligands, as this process entails the incorporation of diverse functional groups onto its surface [40]. The chemical modification of biochar may be executed through various methods, including acid/alkali treatment, oxidation, and co-polymerization grafting [36].

In the current study, biochar was prepared by the low-temperature pyrolysis of rice husk. The resulting biochar was then subjected to chemical modification with 1-[3-(trimethoxysilyl) propyl] urea, which served to introduce an amide group onto its surface. The amide-modified biochar was characterized by scanning electron microscopy (SEM), FTIR, XRD, and thermo gravimetric analysis (TGA). The amide-modified biochar was utilized for the adsorptive removal of chromium VI from wastewater. Batch experiments were conducted to investigate the influence of metal concentration, adsorbent dosage, contact time, and pH values on the adsorption of Cr(VI). The Freundlich, Langmuir, and Temkin isotherm models were utilized in the analysis of experimental results in order to determine the optimal fit for the adsorption process of Cr(VI) onto ABC. Several kinetic models were examined to comprehend the interaction mechanism between chromium (VI) and biochar. The biochar was regenerated and reused several times for chromium (VI) removal from wastewater, and a discernible decrease in adsorption efficacy was observed.

## 2. Results and Discussion

### 2.1. Morphology of the Raw and Amide-Modified Biochar

Figure 1A,B display high-magnification SEM images obtained through scanning electron microscopy (SEM) of ABC before the uptake of Cr(VI) ions. Figure 1C,D show SEM images of ABC after the adsorption of Cr(VI). Figure 1A depicts the existence of openings on the ABC surface, whereas Figure 1C demonstrates the changes in surface morphology after Cr(VI) adsorption. It was observed that the surface structure of ABC underwent alterations after the adsorption of Cr(VI) ions, as indicated by the roughened appearance visible in Figure 1C,D.

### 2.2. FTIR Analysis

The Fourier transform infrared (FTIR) spectra provides insight into the composition of functional groups present on the surface of material. Figure 2 shows the FTIR spectra of rice husk biomass (A) and amide-modified biochar (B). In Figure 2A, the peaks in the wave numbers 3391 cm^−1^ and 2870 cm^−1^ are characteristic of the stretching vibration of O-–H and C–H bonds in the cellulose of rice husk biomass. The peaks in the same range can be seen in Figure 2B, but the intensities are very low, which means that during biochar formation and chemical modifications, these groups have been modified. The band at 2900 cm^−1^ is attributed to the CH stretching vibration of all hydrocarbon constituents in a material. A small peak located at 1690 cm^−1^ in Figure 2A corresponds to the vibration of water molecules absorbed in cellulose biomass. A strong peak at 1656 cm^−1^ in Figure 2B represents amide (–CONH_2_) groups. The peak at 1510 cm^−1^ observed was due to the asymmetrical stretching of carboxyl (–COO−) groups [41], while such peaks were absent in unmodified rice husk biomass. The FTIR spectra clarified the chemical modification of biochar and showed the presence of amide groups [42].

### 2.3. XRD Analysis

The XRD spectra of biochar before and after Cr(VI) adsorption is shown in Appendix A, respectively. XRD spectra of biochar before adsorption is quite smooth and there is no prominent peak, as can be seen in Appendix A, which shows the amorphous nature of the biochar. However, the XRD spectrum of biochar after Cr(VI) adsorption shows two peaks, i.e., 44.12° and 64.4° at 2θ values. These peaks reflect the chromium transfer from the solution to the adsorbent surface. Furthermore, by comparing these peaks with the inorganic crystalline system data library, it was found that the peaks at 44.12° (2θ value) correspond to the existence of chromium oxide specie, i.e., Cr_2_O_3_ having a tetragonal crystalline phase [43]. The peak at 2θ values of 64.44° confirms the presence of chromium oxide [44].

### 2.4. BET Surface Area, Particle Size, and Pore Size of the Biochar

The BET surface area and pore volume of biochar are shown in Appendix A. The surface areas and pore volumes of biochar were increased with pyrolysis. The surface area and pore volume of biochar were found to be 267.16 m^2^/g and 0.052 cm^3^/g, respectively. On the other hand, the powdered rice husk exhibits a surface area of 17.64 m^2^/g and a pore volume of 0.0037 cm^3^/g. This increase in surface area is due to the removal of carbon mass as volatile matters from the biomass surface, which creates pores in the resulting biochar’s structure. The particle size of biochar is shown in Appendix A. The particle size data corresponding to particle fractions d(0.5) for volume-based particle size distribution is summarized in Appendix A. The average particle sizes of biochar are 2–10 µm. The results in Appendix A show that the particle size of biochar increased a little bit with chemical modification. The results of the biochar particle size in the current study are in close agreement with the work presented in Ref. [45].

### 2.5. The Effect of Concentration, pH, Time, and Dose on Cr(VI) Adsorption onto Amide-Modified Biochar

#### 2.5.1. The Effect of Concentration on Cr(VI) Adsorption

The concentration of metal exerts a significant influence on the adsorption of metals onto the surface of an adsorbent. In order to investigate the potential impact of concentration on the removal of Cr(VI) on ABC, several parameters, including pH, adsorbent dosage, temperature, and time, were held constant. In an experimental setup conforming to standard protocols, a series of five conical flasks, each with a volume of 250 mL, were utilized. In an exacting manner, 2 g of ABC was added to each flask while maintaining a constant water bath temperature at 25 °C. Subsequently, 100 mL of each diluted solution was added to the flask that contained ABC. The pH was precisely set to 2.0, following which the flask containing metal solutions and adsorbents was subjected to vigorous shaking for a period of 2 h. Upon completion of the filtration process, an analytical procedure utilizing atomic absorption spectroscopy (AAS) was used to determine the concentration of Cr(VI) ions present in each filtrate.

Figure 3A illustrates the impact of varying metal ion concentrations on the removal of Cr(VI) by ABC. The illustrated graph in Figure 3 presents findings indicating that the removal of Cr(VI) is at 97% when the concentration level is at 100 ppm, albeit exhibiting a slight decline as the concentration of metal ions is raised. At reduced concentrations, the active site located on the surface of ABC is adequate to capture the metal ions, thereby facilitating their adsorption. As the concentration of metal ions was elevated to 200 ppm, the percentage of removal efficiency experienced a decrement and ultimately reached the lowermost limit of 52% when the concentration was escalated to 800 ppm. The present adsorbent possesses the ability to effectively adsorb a significant quantity of chromium (VI) from a concentrated solution. In the domain of literature, the metal ion concentrations employed in prior research studies generally ranged between 10 and 100 ppm. However, the present investigation successfully extracted 97% and 52% of Cr(VI) from 100 ppm and 800 ppm solutions, respectively (as depicted in Figure 3A). The observed reduction in the removal of metal ions with an increase in concentration could potentially be attributed to the complete saturation of available active sites or functional groups on the adsorbent (ABC) by the metal ions. This phenomenon warrants further investigation to better comprehend the mechanisms underlying this phenomenon. According to the findings depicted in Figure 3A, it can be deduced that ABC demonstrates a notable affinity towards Cr(VI) ions across a range of concentrations. At concentrations of 400–600 parts per million (ppm), the removal rates of chromium (VI) were observed to be 90% and 80%, correspondingly. It is anticipated that the utilization of ABC could serve as a viable method for the remediation of wastewater contaminated with Cr(VI). A similar trend was reported in the literature by different researchers [46,47].

#### 2.5.2. The Effect of Adsorbent Dose on Cr(VI) Adsorption

This study aimed to investigate the impact of varying sorbent dosages on the lowest quantity of ABC required for the efficient elimination of chromium (VI) ions. To achieve this objective, quantities of 0.5–3.5 g ABC were introduced into five separate flasks, each containing a 100 ppm solution of chromium (VI). The pH of the solution was adjusted to 2.0, and the adsorption process was conducted for 1 h, under ambient temperature conditions of 25 °C. The findings depicted in Figure 3B demonstrate the impact of varying ABC doses on the removal of Cr(VI). The findings of this study indicate that the percentage of Cr(VI) removal was merely 50% at a lower adsorbent dose of 0.5 g, but it showed a positive correlation with an increment in adsorbent dose up to 2 g/L (as evidenced in Figure 3). At lower dosages of adsorbent, the uptake of Cr(VI) ions exhibited a diminutive reduction, with a recorded removal rate of 50% observed at an adsorbent concentration of 0.5 g/L. This outcome can be attributed to the inadequacy of the adsorbent mass (i.e., ABC) to adsorb the total population of Cr(VI) ions present within the system. Upon increasing the quantity of adsorbent to 1 g per liter, while maintaining the concentration of Cr(VI) at 400 ppm, a noteworthy surge in % Cr(VI) removal to 70% was observed. In a parallel manner, the quantity of adsorbent was augmented to 2 g/L in the context of a 400 ppm Cr(VI) solution, resulting in a recorded removal percentage of approximately 97% for Cr(VI). The quantity of adsorbent was subsequently augmented to 2.5–3 g/L for identical solution conditions (400 ppm), resulting in a 97% removal of % Cr(VI), as portrayed in Figure 2B. The findings of this study indicate that the augmentation of ABC concentration to levels above 2 g/L in the context of a 400 ppm Cr(VI) solution does not yield any significant improvement beyond 97% in the removal of Cr(VI). The findings indicate that a concentration of 2 g/L of ABC exhibits an adsorption capacity of 400 ppm of chromium (VI) and is deemed the most effective dosage. The number of active sites or functional groups present in two grams of ABC was determined to be sufficient for the adsorption of chromium (VI) from a solution with a concentration of 400 ppm. The results of the adsorbent dose effect on Cr(VI) removal in the current study is in close agreement with the work published in Refs. [48,49].

#### 2.5.3. The Effect of Contact Time on Cr(VI) Adsorption

The adsorption time duration represents a significant parameter in the context of batch adsorption experiments. Figure 3C depicts the impact of time duration on the adsorption of Cr(VI) onto ABC. The experimental conditions remained fixed for other parameters, namely temperature (25 °C), pH (2), agitation speed (300 rpm), Cr(VI) concentration (100 ppm), and ABC quantity (1 g/L). The findings indicate that the removal of %Cr(VI) exhibited an escalating trend until the attainment of equilibrium at the 2 h mark, with a corresponding % removal of 97%. No significant improvement in the elimination of hexavalent chromium (%Cr(VI)) was observed after a time period exceeding 2 h. At the outset, it was observed that there was a positive correlation between the duration of adsorption and the removal of Cr(VI). This is likely because the metal ions exhibited an affinity for the active sites of the amide-modified biochar, thereby occupying them fully within a period of two hours. There was no discernible advancement in %Cr(VI) elimination beyond the 2 h mark as a result of the complete saturation of the reactive sites of ABC with Cr(VI) ions. A similar trend was observed for the removal of Cr(VI) from wastewater in the literature [50,51].

#### 2.5.4. The Effect of pH on Cr(VI) Adsorption

The potential impact of the pH values on the efficiency of Cr(VI) elimination through the use of ABC was subjected to scrutiny, with all other factors remaining unchanged. Figure 3D depicts the impact of pH on the removal of Cr(VI). The analysis conducted reveals that pH 2 was the ideal condition for Cr(VI), with the highest level of elimination noted. At a pH below 3, aqueous solutions contain HCrO_4_^−^, Cr_2_O_7_^2−^ and CrO_4_^2−^ forms of chromium. The surface of ABC also undergoes protonation at low pH, leading to a powerful electrostatic attraction between the positively charged ABC surface and the aforementioned ions. As the pH is raised, the capacity of the surface of ABC to interact with oxyanions of Cr(VI), specifically HCrO^4−^, Cr_2_O_7_^2−^, and CrO_4_^2−^, is diminished. This can be attributed to the surface obtaining a negative charge under such conditions, thereby reducing the attraction between the surface and the oxyanions of Cr(VI) as a consequence. The findings suggest that the relationship between ABC and Cr(VI) oxyanions is influenced by changes in pH [49,52].

### 2.6. Adsorption Isotherms

#### 2.6.1. Freundlich Adsorption Isotherm

The Freundlich isotherm posits that the process of adsorption occurs in multiple layers on a surface that exhibits heterogeneity, with the adsorption rate increasing as concentration levels rise until equilibrium is reached. An identical trend was noted in regard to the absorption of hexavalent chromium onto amide-modified biochar (ABC). The Freundlich adsorption isotherm can be expressed mathematically in its linear form, represented as given in Equation (1) [53].
(1)$$\mathrm{logqe}=\mathrm{logKf}+\frac{1}{n}{\;\mathrm{logC}}_{e}$$

Through the depiction of logarithmic curves plotting ln qe versus ln Ce, linear plots were acquired, as evidenced in Figure 4A, whereby a slope of 1/n was observed. The values for n and Kf (L mg^−1^), denoting the adsorption capacity, were determined through graphical analysis. The results indicate that the adsorbent surface exhibits a highly favorable interaction with Cr(VI) ions, as evidenced by the “n” values [54,55].

#### 2.6.2. Langmuir Adsorption Isotherm

The linear form of Langmuir adsorption isotherm can be written as given in Equation (2) [56],
(2)$$\;\frac{C_{e}}{q}=\frac{1}{q_{\max}k}+\frac{C_{e}}{q_{\max}}$$

“Ce” is metal ions concentration at equilibrium, qe (mgg^−1^) is the number of metal ions adsorbed per unit mass of adsorbent, and KL(L mg^−1^) is the Langmuir isotherm constant. These parameters are pertinent to the analysis of the adsorption process under examination [56], plotting the Ce/q against Ce data for Cr(VI) adsorption, resulting in the formation of a linear relationship with a slope of 1/q_max_ and an intercept of 1/q_max_k. The equilibrium parameter (RL) was ascertained through the utilization of Equation (3).
(3)$$R_{L}=\frac{1}{1+K_{L}C_{0}}$$

The findings presented in Table 1 indicate that the isotherm constants and regression values are indicative of a strong fit between the adsorption data of Cr(VI) onto ABC and the Langmuir isotherm model. Furthermore, the RL values are lower than one. The Langmuir isotherm model for the removal of Cr(VI) using ABC is depicted in Figure 4B. The results presented here are in close agreement with those reported in the literature [57,58].

#### 2.6.3. Temkin Adsorption Isotherm

The Temkin adsorption isotherm presents a valuable means by which to elucidate the interactions between adsorbent and adsorbate, as well as to approximate the enthalpy of adsorption. The heat of adsorption (B_T_) may be computed through Equation (4)
(4)$$\mathrm{qe}=\frac{R_{T}}{b_{T}}\;\left({\mathrm{lnK}}_{T}C_{e}\right)=B_{T}\left({\mathrm{lnK}}_{T}C_{e}\right)$$

In this equation, B_T_ represents the adsorption heat, and T is the absolute temperature in units of Kelvin. Additionally, b_T_ is the Temkin isotherm constant expressed in units of J/mol. This constant denotes the variation of adsorption energy and represents the equilibrium binding constant that corresponds to the maximum binding energy. Both B_T_ and K_T_ can be determined through the derivation of the slope and the intercept of a linear plot of qe versus lnCe. The Temkin model postulates a linear reduction in the heat of adsorption of molecules in the layer with an increase in coverage. The adsorption of Cr(VI) was analyzed by plotting the quantity of adsorbed Cr(VI) (qe) against the natural logarithm of the equilibrium concentration (lnCe), as depicted in Figure 4C. The Temkin isotherm constants were determined using the slope and intercept of the qe versus lnCe plot. The observed increase in the binding energy (AT) of amide-modified biochar as the temperature rises suggests a strong influence of thermal properties on the system. The interaction between Cr(VI) ions and the adsorption sites displayed an increasing trend with rising temperature, thereby indicating the endothermic nature of the adsorption of Cr(VI) on ABC. This finding has been listed in Table 1 for further reference. The thermodynamic binding constant (B_T_) for amide-modified biochar (ABC) is found to be 54.36 kJ mol^−1^, indicating a strong association between chromium (VI) ions and ABC. The correlation coefficient ascertained for the Temkin isotherm indicates a significant correlation, thereby substantiating the superior fitting of the Temkin model with respect to the experimental data pertaining to the adsorption of Cr(VI) onto ABC [59,60].

### 2.7. Adsorption Kinetics

A study on kinetics was conducted with the objective of determining the mechanism and rate-determining step involved in the adsorption of chromium (VI) onto ABC. To determine the ideal time for achieving maximum adsorption of Cr(VI), a series of batch adsorption experiments were conducted covering various time intervals, with the level of adsorption at specific time intervals (qt) being subsequently determined. The process of Cr(VI) adsorption onto ABC exhibited initial rapid kinetics until the 60th minute, followed by slower kinetics until the attainment of equilibrium. The findings indicate that the adsorption of Cr(VI) onto ABC is categorized as chemisorption, and the metallic ions engage with diverse functional groups of ABC. The present study employed pseudo-first-order, pseudo-second-order, and intra-particle diffusion models for the purpose of identifying the rate-limiting step.

#### 2.7.1. Pseudo First-Order Kinetic Model

The linear form of pseudo-first-order reaction can be written as in Equation (5),
(5)$$\ln\left(q_{e}-{\;q}_{t}\right)={\mathrm{lnq}}_{e}-k_{1}t$$

The quantity of Cr(VI) adsorbed on ABC at equilibrium is referred to as q_e_ (mg/g), whereas the amount adsorbed on ABC at a specific time (t) is represented by the variable q_t_ (mg/g). The rate constant, expressed in terms of minutes, is denoted as k_1_. The graph depicting the relationship between the natural logarithm of the quantity of Cr(VI) remaining in solution (q_e_) at equilibrium and the quantity removed at a specific time (q_t_) may be referred to as a plot of ln(q_e_ − q_t_) versus time (t). At different concentrations, a linear relationship is observable, as depicted in Figure 4A. The rate constant for the process, denoted as k_1_ (min^−1^), was determined by evaluating the slope corresponding to the natural logarithm of the difference between the equilibrium concentration (q_e_) and the concentration at any given time (q_t_). The graph illustrates the relationship between two variables through a series of plotted points (Figure 5A). Table 2 presents the outcomes obtained from both experimental and calculated evaluations of q_e_ (mg/g), R_2_, K_1_, and Ki (mgg^−1^ min^−1^), with regard to the pseudo-first-order, pseudo-second-order, and intra-particle diffusion model, in relation to the adsorption of Cr(VI) on ABC.

#### 2.7.2. Pseudo Second-Order Model

The pseudo-second-order kinetic model can be written in linear form as given in Equation (6),
(6)$$\frac{t}{q^{t}}=\frac{1}{k_{2}q_{e}^{2}}+\frac{t}{q_{e}}$$
where q_e_ and q^t^ are utilized as measures of the quantity of Cr(VI) adsorbed (mg/g) at equilibrium time and specific time (t), respectively. Furthermore, the rate constant k_2_ (g·mg^−1^·min^−1^) is determined in order to analyze the adsorption kinetics. In the context of a linear graph (Figure 5B), the adsorption capacity (q_e_) and rate constant k_2_ (g·mg^−1^·min^−1^) were determined through calculation. The findings demonstrate that the second-order model is a more appropriate choice than the first-order model to characterize the adsorption kinetics of chromium (VI) onto ABC.

#### 2.7.3. Intra-Particles Diffusion Model

An intra-particle diffusion model was employed to examine the intricate process of diffusion of hexavalent chromium onto ABC, as documented in Ref. [52]. The phenomenon of intra-particle diffusion can be mathematically expressed through the following Equation (7).
(7)$$q_{t}={\;K}_{p}t^{0.5}+C$$

The factor qt (mg) can be observed in the given measurements; qt (mg·g^−1^) is the concentration of Cr(VI) adsorbed onto the adsorbent at time t. The variable C denotes the intercept, while Kp (mg·g^−1^·min^0.5^) signifies the intra-particle diffusion constant. The thickness of the boundary layer exhibits a direct relationship with the intercept, whereby a greater intercept connotes a more pronounced boundary layer effect, and conversely, a lower intercept indicates a reduced boundary layer effect. This finding is supported by prior research [53]. It is observed from Table 2 that the intercept value is elevated, indicating that the deposition of Cr(VI) ions onto ABC results in a substantial layer formed over the surface of ABC. This suggests that boundary layer diffusion governs the rate-controlling step for the adsorption of Cr onto ABC, which is dominant at high concentrations of Cr(VI). The observation substantiates the presence of a distinct boundary layer phenomenon along the surface of ABC. All three kinetic models are deemed appropriate for the phenomenon of Cr(VI) adsorption onto ABC, as exemplified by the linear nature of the curve presented (refer to Figure 5C). The results obtained from the calculation of qe exhibit a high degree of proximity to the corresponding experimental values. Additionally, the R-squared value for the pseudo-second-order kinetic model is reported as 0.99, as presented in Table 2. Of the three kinetics models considered, the pseudo-second-order model exhibits a higher degree of correlation with the experimental data for Cr adsorption onto ABC, followed by intra-particle diffusion.

### 2.8. Regeneration and Reuse of the ABC and Cr(VI) Recovery

To regenerate the used adsorbent (ABC), a sample of ABC imbued with adsorbed chromium (VI) ions was subjected to stirring in a flask containing 100 mL each of 0.5 M nitric acid and 0.4 M ethylenediaminetetraacetic acid, individually. The mixture consisting of previously utilized adsorbent (ABC) was agitated at ambient conditions for a duration of 3 h, subsequently subject to filtration, and the extracted metal in the filtrate was measured via atomic absorption spectroscopy. The percentage of desorption of chromium (VI) ions was computed using the following equation (Equation (8)).
(8)$$\%\;\mathrm{Desorption}=\frac{\mathrm{amount}\;\mathrm{of}\;\mathrm{Cr}\left(\mathrm{VI}\right)\;\mathrm{desorbed}}{\mathrm{amount}\;\mathrm{of}\;\mathrm{Cr}\left(\mathrm{VI}\right)\;\mathrm{adsorbed}}\times 100$$

In order to assess the viability of utilizing ABC sorbent for multiple cycles of metal ion sequestration, a series of five regeneration experiments were conducted. The regenerated ABC was recovered using various solvents, including sodium hydroxide (NaOH), sodium chloride (NaCl), and distilled water, in order to assess their effectiveness as regeneration mediums. The regenerated ABC was subsequently employed in sequential cycles of Cr(VI) ion sequestration [61].

The findings pertaining to the reusability of ABC biosorbent are delineated in Figure 6. Throughout the five experimental cycles, it was observed that the NaOH solution exhibited the highest degree of ABC regenerability when compared to NaCl and distilled water. The use of regenerant NaOH yielded a potent alkaline pH milieu, which enabled the exchange of Cr(VI) ions with hydroxyl ions, thus promoting the efficient desorption of Cr(VI) ions from the surface of ABC [62]. Through a maximum of three recycling sequences wherein NaOH solution was utilized for regeneration, the ABC biomass revealed a reduction of below five percent in its efficacy for Cr(VI) ions. The efficacy of Cr(VI) adsorption exhibited a reduction exceeding 4% upon the initial NaCl-assisted recycling of the regenerated ABC biosorbent. In the initial cycle run of distilled water, the loss exceeded 8%. In conclusion, the results of this study suggest that ABC possesses a considerable degree of regenerability potential, indicating that it can continue to be utilized as an effective for removing Cr(VI) ions from wastewater.

The desorption efficiency of ABC is influenced by the concentration of the regenerating agent and the duration of the desorption process. The regenerative agent NaOH was utilized during a desorption interval of 1, at a concentration of 0.1 M. Applying moderate desorption settings was crucial to guarantee a successful extraction process without damaging the functional surface of ABC. Optimizing the parameters may lead to an increase in the recovery rate of the ABC. To facilitate the disposal of ABC laden with Cr(VI) after undergoing five cycles of recycling, it is necessary to initiate an exhaustive chromium recovery phase from the biomass prior to its elimination. Such an approach will effectively mitigate the environmental impact of the discarded biomass. During this stage, the adsorbed chromium is subjected to hydrolytic precipitation in the presence of 1 M NaOH, resulting in the formation of chromium hydroxide (Cr(OH)3). This process is facilitated by a prolonged desorption duration. The compound Cr_2_O_3_ can be readily solubilized in acid solutions, such as H_2_SO_4_, and employed as both a tanning agent and a starting material for the synthesis of a wide range of chromium-based compounds [62].

### 2.9. Comparison of Cr(VI) Removal by ABC Prepared in This Study with Other Biochar-Based Adsorbents

The performance of ABC developed in the current study was compared with other adsorbents for the removal of Cr, and the findings are tabulated in Table 3. Furthermore, a comprehensive evaluation is conducted in comparison to other biochar-based adsorbents (Table 3). According to a prior study, Tobacco petiole pyrolytic biochar has been found to possess the highest adsorption capacity, standing at 195.2 mg/g [63]. However, the present study has reported the adsorption capacity of amide-modified biochar derived from rice husk biomass as 215.42 mg/g. No prior studies have reported biochar exhibiting such a significant capacity for the removal of Cr(VI) through adsorption.

## 3. Materials and Methods

Acetic acid (>99%), hydrochloric acid (37%), 1-[3-(trimethoxysilyl) propyl] urea (97%), ferric sulfate (97%), hydrogen peroxide (35%), and NaOH (>98%), were bought from Sigma–Aldrich, Schnelldorf, Germany. A standardized solution of chromium (VI) with a concentration of 1000 parts per million (ppm) was diluted to prepare solutions with concentrations of 100, 200, 300, 500, 600, 700, and 800 ppm. Wastewater samples were collected from Pakistan Steel Re-Rolling Mills, Hattar industrial area, Haripur, Pakistan.

### 3.1. Biochar Preparation

The rice husks were obtained from a local farm situated in Swat, Khyber Pakhtunkhwa, Pakistan. The rice husks were subjected to washing and cutting, followed by drying in an oven at 70 °C for 5 h. The dried rice husks were pyrolyzed in a tubing furnace for 3 h at a temperature of 400 °C. After three hours, the biochar was taken out of the furnace and kept in a desiccator at room temperature.

### 3.2. Chemical Modification of Biochar

Raw biochar (100 g) was put in a round bottom flask of 500 mL capacity, and 2 % HCl solution (300 mL) was added. The mixture was then heated at a temperature of 80 °C for a period of 3 h. After acid hydrolysis, biochar was strained using cotton bags, rinsed with deionized water until it achieved a neutral pH, and placed in an oven at 70 °C for 6 h. Then, 100 g of acid-hydrolyzed biochar was dispersed in a mixture of 0.2 mL of hydrogen peroxide, 2 mL of 1-[3-(trimethoxysilyl) propyl] urea, and 0.3 mL of acetic acid in a conical flask. The flask was then placed in a water bath for 3 h at a temperature of 80 °C. The contents were filtered, cleansed with deionized water, and subjected to oven drying at 70 °C for a duration of 5 h.

### 3.3. Characterization of Biochar

The biochar’s composition was ascertained by means of elemental analysis through an Elemental Analyzer (NA-1500, Carlo-Erba). The BELSORP-Max (Osaka, Japan) was used to measure the surface area, pore volume, and pore size of ABC. The pore volume of ABC was measured from the quantity of nitrogen gas absorbed at a particular pressure. The surface morphology of the biochar was evaluated with the S-4200 FE-SEM instrument made by HITACHI, Tokyo, Japan. The Fourier transform infrared spectroscopy (FTIR) was utilized to examine the surface functional groups of biochar. The analysis was conducted using the TENSOR II machine from Bruker, Billerica, MA, USA. The concentration of Cr(VI) in the filtrate was determined by an atomic absorption spectrometer using a 240 FS atomic absorption spectrometer (Agilent), Santa Carla, CA, USA.

### 3.4. Adsorption of Chromium VI on Amide-Modified Biochar (ABC)

The adsorption of chromium VI onto amide-modified biochar was checked in batch experiments. To find out the optimum values of parameters such as pH, adsorbent dose, adsorption time, metal concentration, and temperature on Cr(VI) adsorption onto amide-modified biochar, batch adsorption experiments were conducted. A stock solution of chromium (500 ppm) was prepared and diluted into different concentrations (100–500 ppm), and these diluted solutions were used for further experiments. Then, 200 mL of each solution was added into a flask of 500 mL capacity, 0.1 g adsorbent (ABC) was added into it, and the flask was shaken at 25 °C for one hour in a water bath. After 1 h, the content was filtered, and the concentration of hexavalent chromium in the filtrate was determined using an atomic absorption spectrometer (Perkin Elmer AAnalyst 800 Model FAAS), Waltham, MA, USA, equipped with a 10 cm air/acetylene burner and a chromium hollow cathode lamp. The lamp current was 12 mA, the wavelength was 357.9 nm, the fuel flow rate was 1.5 L/min, the bandpass was 0.7 nm, and the sample suction rate was L1.9 mL/min. The adsorption capacity and % removal of chromium was calculated using Equations (9) and (10), respectively.
(9)$$\mathrm{qe}=\frac{\left(\mathrm{Co}-\mathrm{Ce}\right)V}{m}$$
(10)$$\mathrm{Cr}\left(\mathrm{VI}\right)\mathrm{removal}\;\left(\%\right)=\frac{\left(\mathrm{Co}-\mathrm{Ce}\right)}{\mathrm{Ce}}\times 100$$
where qe is adsorption capacity at equilibrium time, V is solution volume in a liter, m is the mass of adsorbent in grams, and “Co” and “Ce” are metal concentrations before and after adsorption. Wastewater was collected from Hattar Steel Re-Rolling Mills, Hattar industrial estate, Haripur, KP, Pakistan. The physicochemical properties of wastewater, such as pH, salinity, total dissolved solids, biological oxygen demand, chemical oxygen demand, conductivity, and hardness, are summarized in Appendix A.

### 3.5. Adsorption Isotherms and Kinetics

An investigation of the adsorption isotherm was conducted in order to elucidate the mechanism underlying the transfer of metal ions from the solution to the adsorbent. The impact of concentration (Ce) on the adsorption capacity (qe) under constant temperature was measured in batch adsorption experiments. The isotherm study was carried out keeping the optimum pH (2), adsorbent dose (3 g/L), and time (60 min), while the kinetics study was also performed under optimum conditions (pH, dose, etc.) while changing the adsorption time. The adsorption data were computed by Langmuir, Freundlich, and Temkin isotherm models to check the fitness of adsorption data with these isotherm models. The data computation and linear regression analysis were performed utilizing the Origin Pro-8 software. Furthermore, an investigation into the kinetics of adsorption was conducted with the aim of determining the rate of transfer of the adsorbate from the solution onto the surface of the adsorbent material, as well as the time that the adsorbate remained on the adsorbent surface. Several kinetics models, including the pseudo 1st order, pseudo 2nd order, and intra-particle diffusion model, were utilized to compute the adsorption data of Cr(VI) onto ABC.

### 3.6. Reproducibility of ABC for Cr(VI) Adsorption

In order to assess the efficacy of multiple batches of ABC with regard to Cr(VI) adsorption, three separate batches were synthesized utilizing identical methods of preparation. The aforementioned batches were subsequently employed to adsorb Cr(VI) utilizing equivalent optimized conditions with respect to pH, dose, time, and concentration. The adsorption efficacy of every batch of ABC for Cr(VI) removal was computed, and the results of each batch were compared. The findings indicate that the adsorption efficiency exhibited similar values across all three batches. The relative standard deviation of adsorption capacity for three batches was noted to be 0.8%. A calibration curve was constructed for the Cr(VI) solution, and the blank samples were utilized to assess any potential errors in the measurement. Marquardt’s %RSD was employed as an indicator of error in formulating calculations for the adsorption capacity of Cr(VI), as specified in Equation (11).
(11)$$\overset{N}{\underset{i=1}{\sum }}{\left[\frac{q_{e\left(\exp\right)}-q_{e\left(\mathrm{cal}\right)}}{q_{e\left(\exp\right)}}\right]}_{i}^{2}$$

### 3.7. Regeneration

The recycling of adsorbents plays a significant role in influencing the economic viability of the adsorption process, as well as its associated operational costs [45]. In order to regenerate the utilized adsorbent, 1 g of used ABC was subjected to treatment with a desorbing solution consisting of 0.1 M NaOH, 0.1 M NaCl, and distilled water. The mixture was agitated for 1 h at a temperature of 25 °C at a rotational speed of 100 revolutions per minute. Before employing the regenerated ABC for Cr(VI) adsorption, it underwent multiple rinses using distilled water to ensure the removal of any contaminants. Following final rinsing, the ABC was subsequently dried and stored for reuse.

## 4. Conclusions

The present study describes the utilization of rice husk for the preparation of amide-modified biochar through the process of low-temperature pyrolysis. This modified biochar was utilized for the removal of Cr(VI) ions from wastewater. This study involved the treatment of biochar derived from rice husk biomass with a dilute hydrochloric acid solution, followed by subsequent treatment with hydrogen peroxide for the purpose of activating and oxidizing the surface functional groups of the biochar. The biochar that was produced underwent characterization by using scanning electron microscopy and thermo gravimetric analysis. This study revealed that the amide-modified biochar exhibited a significant removal efficiency of 97% for Cr(VI). The exceptional adsorption capacity observed in amide-modified biochar can be attributed to their porous nature and the presence of surface functional groups incorporated through chemical modification. The findings concerning the adsorption of Cr(VI) onto CMB were assessed through the application of the Friundlich, Langmuir, and Temkin isotherm models. The findings indicate that the utilization of CMB as an adsorbent exhibits promising potential for the efficient removal of Cr(VI) and other heavy metals from wastewater owing to its high removal efficiency and cost-effectiveness. In the foreseeable future, amide-modified biochar is expected to serve as a valuable tool for the removal of various heavy metals and dyes present in wastewater in fixed bed columns. The amide-modified biochar will be checked for multi-step adsorption in future work. Detail mechanism of Cr(VI) adsorption onto ABC will also be discussed in our future work.

## Figures and Tables

**Figure 1 molecules-28-05146-f001:**
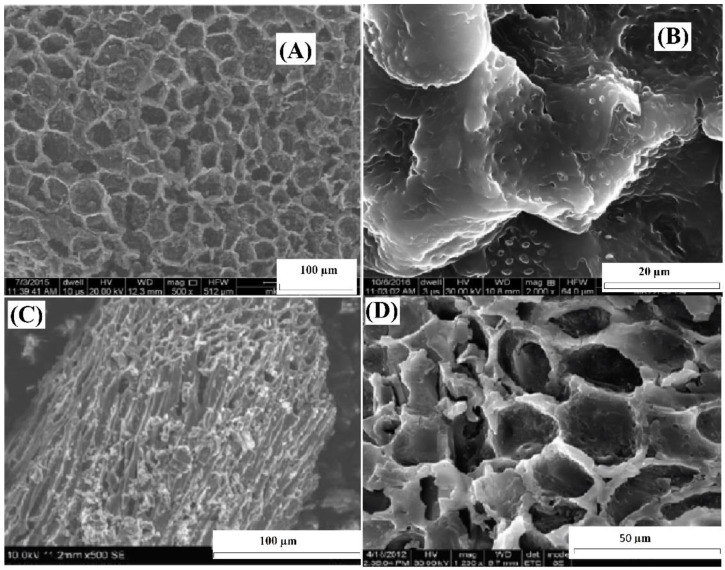
SEM images of ABC (**A**), close view of ABC before adsorption (**B**), close view of ABC after Cr(VI) adsorption (**C**), and closer view of ABC after Cr(VI) adsorption (**D**).

**Figure 2 molecules-28-05146-f002:**
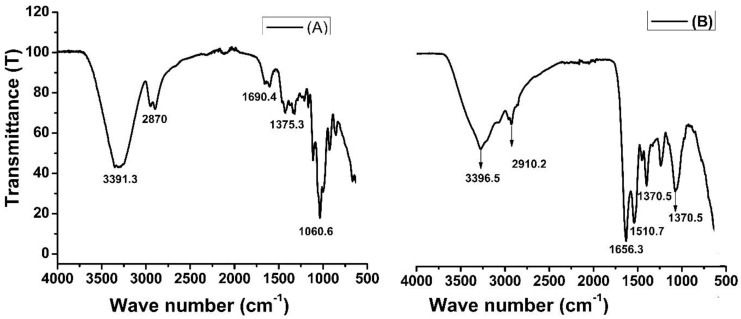
FTIR spectra of rice husk biomass (**A**) and amide-modified biochar (**B**).

**Figure 3 molecules-28-05146-f003:**
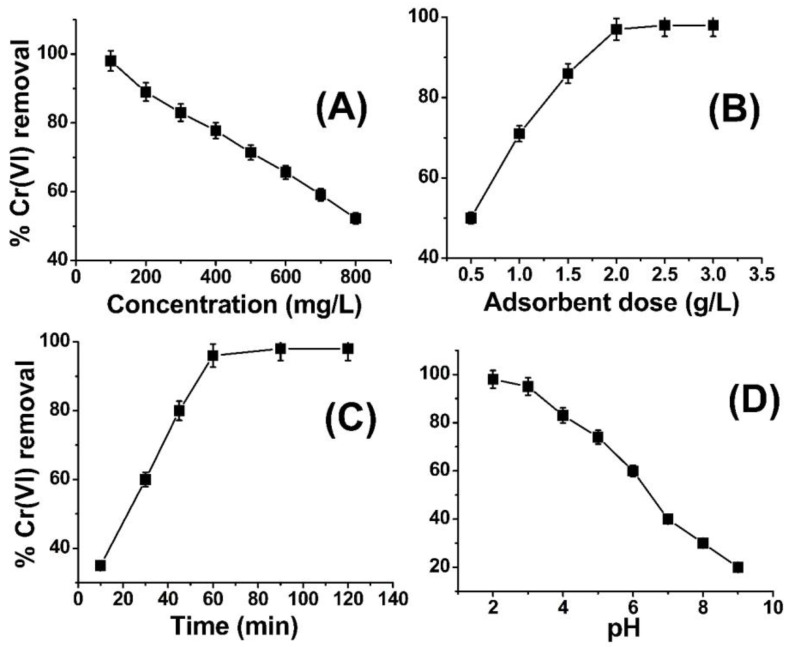
Effect of concentration (**A**), adsorbent dose (**B**), contact time (**C**), and pH (**D**) on Cr(VI) removal using amide-modified biochar.

**Figure 4 molecules-28-05146-f004:**
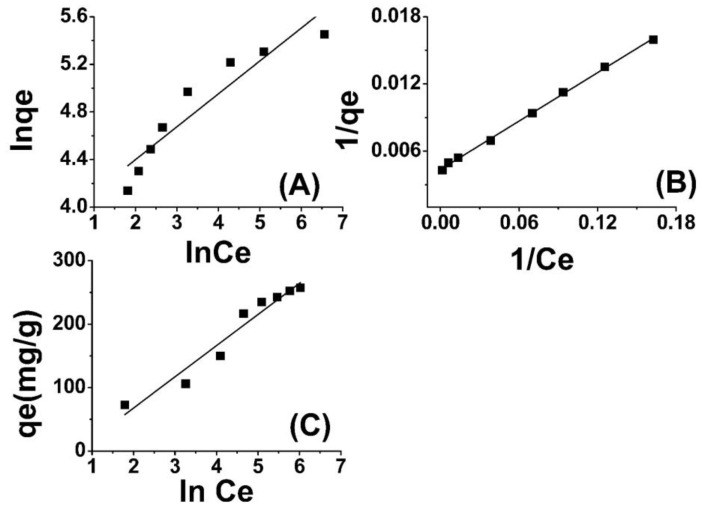
Freundlich adsorption isotherm model (**A**), Langmuir model (**B**), and Temkin model (**C**) for Cr(VI) adsorption onto amide-modified biochar.

**Figure 5 molecules-28-05146-f005:**
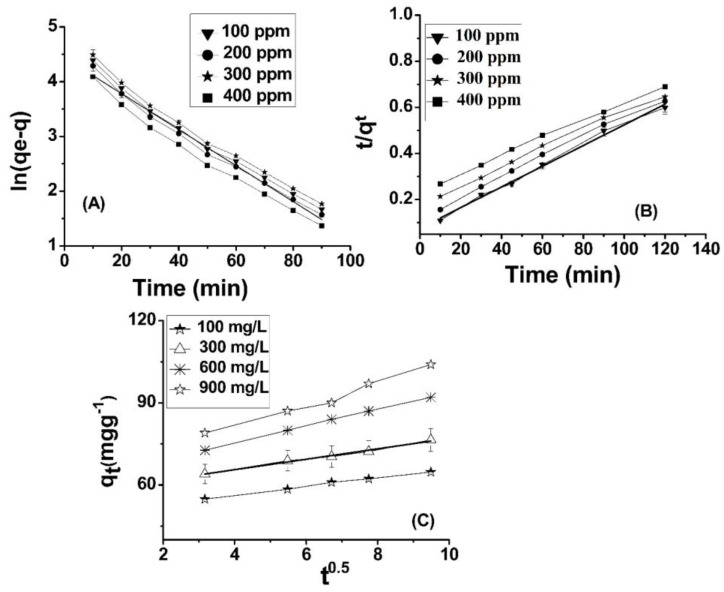
Pseudo-first-order (**A**), pseudo-second-order (**B**), and intra-particle diffusion plots (**C**) for Pb(II) adsorption onto ABC.

**Figure 6 molecules-28-05146-f006:**
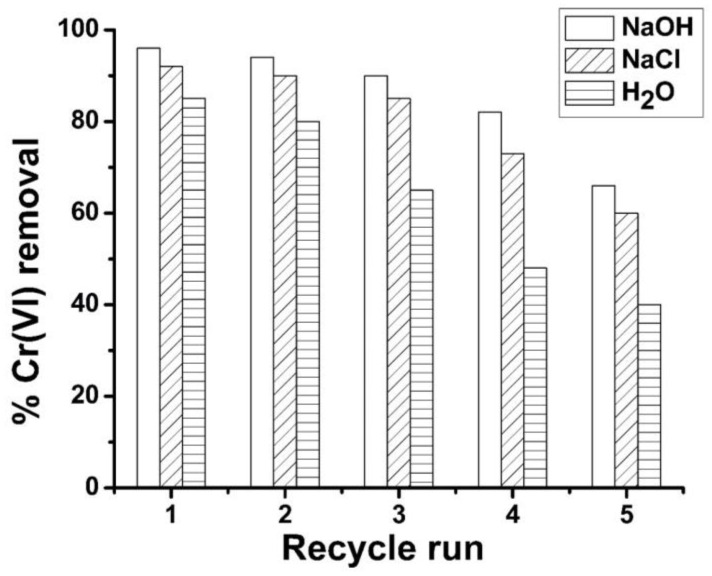
Regeneration and reuse of ABC for Cr(VI) removal.

**Table 1 molecules-28-05146-t001:** Adsorption isotherms parameters for Cr(VI) adsorption onto amide-modified biochar.

Isotherm	Adsorption Isotherm Parameters	
Freundlich	1/n	3.6166
	K_F_ (mg/g)	46.758
	R^2^	0.940
Langmuir	q_max_ (mg/g)	229.88
	K (L/mg)	0.0730
	R^2^	0.999
Temkin	B_T_ (KJmol^−1^)	49.117
	K_T_ (Lmg^−1^)	0.5385
	R^2^	0.963

**Table 2 molecules-28-05146-t002:** Pseudo-first-order (calculated values), pseudo-second-order (calculated), and intra-particle diffusion values (calculated) for Cr(VI) adsorption onto ABC.

C_0_ (mg/L)	Pseudo 1st Order Calculated		Pseudo 2nd Order Calculated			Intra-Particle Diffusion Values Calculated	
	qe	R^2^	qe (mg/g)	K_2_	R^2^	R^2^	K_i_ (mgg^−1^ min^−1^)
400	173.67	0.9493	165.34	0.0543	0.998	0.9343	3.35
500	296.56	0.9426	322.12	0.0456	0.997	0.9278	2.34
600	337.34	0.9578	387.30	0.0367	0.999	0.9545	4.56
700	345.21	0.9598	392.71	0.0334	0.995	0.9425	4.56

**Table 3 molecules-28-05146-t003:** Comparison between ABC and other adsorbents for Cr(VI) removal.

Adsorbent	q_max_ (mg/g)	Ref
Magnetic bagasse BC	29.08	[64]
Municipal sludge BC	20	[65]
*Acorus calamus Linn* BC	46.65	[66]
Feather waste BC	56.64	[66]
Tobacco petiole BC	195.2	[67]
Enteromorpha prolifera BC	88.17	[68]
*Camellia oleifera* shells BC	54.38	[69]
Raw corncob BC	25.94	[62]
Amide-modified biochar	215.42	Current study

## Data Availability

The data will be provided by the corresponding author upon written request.

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
