# Peer review of "Efficient Removal of Hexavalent Chromium (Cr(VI)) from Wastewater Using Amide-Modified Biochar"

_molecules, 2023, doi:10.3390/molecules28135146_

Round 1

Reviewer 1 Report

I appreciate the opportunity to review the manuscript titled:

Efficient removal of Cr (VI) from wastewater using chemically modified biochar

I present my comments on the manuscript below and hope they will be useful to improve the manuscript.

In general, the format of the manuscript should be revised. It also sometimes contains the numbers very close to the units and sometimes does not. The equation format changes constantly.

Not even the references are according to the format required by the journal.

In the abstract and during the materials and methods section, the manuscript mentions that FTIR, XRD, and TGA results will be shown, in addition to the characterization of the specific surface area and pore size of the CBC. However, these results are not reported.

I suggest that for a better understanding, the explanation of the conditions in which the experiments were carried out be placed in the material and methods section, and not in the results and discussion section.

On the other hand, the results of the tables and figures do not show error bars or standard errors.

Line 55

Are the authors referring to Cr(VI)?

Lines 57 y 68

The authors do not previously mention the symbol of Chromium as Cr. They refer to it several times only as Cr, but in line 68, they refer to Cr as chromium. Please correct.

Lines 170-181

Under what conditions were the isotherms and kinetics performed at different initial metal concentrations? At pH 3, or the best pH of 2? At what dose of adsorbent? For how long?

Figure 1

The images do not show the number of magnifications of each one. The scale is not there either. So, the reader's appreciation could be wrong when comparing different biochar increases.

Especially in Figures 1A and 1C, they should compare the adsorbents before and after sorption, as well as 1B and 1D.

Lines 223-234

This part should be in the material and methods section.

Figure 2

The results are expressed as adsorption percentages. However, would the trends change if they were expressed in adsorption capacity (q)? Could the results be graphed according to sorption capacity?

Lines 370-372

The authors mention that the value of BT shows that there is a robust association between Cr(VI) and CBC. What does it mean to be robust? Which values ​​are considered a robust association, and which are not?

Figure 4

Where are the lines of each concentration?

Table 2

Where it says Pseudo 1st order experimental

Does it only correspond to the experimental values ​​of qe?

While K1, qe, and R2 already belong to the values ​​predicted by the pseudo-first-order model? Please correct the headings in the table. 

On the other hand, when C0= 100 mg/L, the qe-experimental is incorrect since the values ​​of qe-exp (235 mg/g) are much higher than the initial metal concentration tested. The maximal obtained qe should be 200 mg/g (if the adsorbent dose was 0.5 g/L). Then, it could not give the reported values ​​of qe-exp. Or what was the dose of the adsorbent?

Verify these data and those in Table 2.

Lines 453-463

This part should be in the material and methods section.

Lines 454-455

I don't understand the function of the acids that are mentioned there, since only the saturated adsorbent should have been brought into contact with the individual solutions of NaOH, NaCl, or water.

Section 3.7 Recovery cycles. 

Could the authors put a figure showing the adsorption presented by the CBC in each cycle to see clearly how the CBC is decreasing, not only its ability to adsorb but also to desorb?

On the other hand, the authors previously mentioned that the association of CBC with Cr(VI) is robust; however, water can remove a high percentage of Cr(VI) (>40%). So, the association presented by CBC-Cr(VI), is it a robust one?

Conclusions

I suggest to the authors that the conclusions provide more compelling data on their findings.

Author Response

Point by point reply to reviewer's comments are appended in pdf file attached here and the changes has been made in the revised manuscript accordingly

Reviewer 2 Report

The authors investigate the removal of Cr(VI) from wastewater using chemically modified biochar (CBC). They characterise the material by SEM and a series of experiments that examine the factors affecting adsorption and desorption of Cr(VI) by the CBC.  The results are interesting and show the potential of CBC as a low cost agent for the removal of Cr(VI).

However, there are some inconsistencies in the paper, some of the methodology is not included and the authors are ascribing significance to differences that are surely within error. 

Specifically:

Line 160: how was  the concentration of hexavalent chromium in the filtrate determined? A description must be added to section 2.4, as this is a crucial measurement.

While Figure 2a,b,c show clear trends, the data is not consistent between them. Thus 2a uses 1 g/l CBC for 2 hours and at 400 ppm Cr(VI) they obtain ~80% removal. In 2b, 1 g/l CBC removes  ~70% of a 400 ppm Cr(VI) solution in 2 hours. In 2c, 1 g/l CBC removes 97% of a 400 ppm solution in 2 hours. The authors must explain these contradictions.

The quality of the fits in Figure 3 are all better than 0.99, so all three models describe the data equally well, so they cannot be discriminated between. What does this mean for the adsorption process?

In Table 2, the authors state that because the R2 values are slightly better by  a tiny amount, the pseudo 2nd order equation better describes the kinetics. The differences are so small that this statement must be justified. Why would they expect 2nd order kinetics to be relevant? What are the error bars on the measurements?

Overall, I believe that this paper is publishable in Molecules, but it requires major revision before this can happen.

Minor points

General point: the convention is that numbers and their units are separated by a space. Please follow this convention.

General point: in English, the names of chemicals and the elements are nouns, not proper nouns, so should not be capitalised, unless they are the first word in a sentence.

General point: please ensure that all the terms in the equations are defined.

General point: error bars should be included in Figures 2-5 and Tables 1-3. If it is not possible to provide them for every value, at least an estimate must be provided.

Line 55: Should Cr(V) be Cr(VI)?

Figure 1: The scale bars are too small to see in the images. Please include a scale bar that is easily visible.

Figure 2A and B: The units of the axes must be included.

Line 394: The natural logarithm should be written as "ln" (all lower case) not "In".

Line 412: In Eqn.(8), is K2 a rate constant (as implied in the following paragraph) or is it an equilibrium constant?

Line 427: What is meant by " the variable g−1 represents"?

Line 494: The 3 in Cr(OH)3 should be a subscript.

Line 509: This should be Table 3 not Table 4.

The standard of English is acceptable, but the paper would benefit from the assistance of an English language editing service.

I am a native English speaker.

Author Response

Point by point response to reviewer's comments are appended in pdf file attached here and the suggested changes has been carried out in the revised manuscript 

Reviewer 3 Report

The paper molecules-2425489, proposes the preparation and the characterization of biochar derived from agricultural waste. Materials was used to eliminate Cr (VI) from wastewater. The effect of some parameters was discussed. Differenta models were tested in addition to the kinetic and isotherm study.

The work was interesting and having significant advancement. The results have scientific relevance, but some point should be reviewed.

Therefore, I recommend the manuscript for publication in “Molecules” after Major revision.

Abstract:

·       More numerical form of results should be mentioned in the abstract.

Introduction

The authors should cite recently reported work on the adsorption of Cr and on the activation/use of Biochar in water treatment, you can cite :

https://doi.org/10.1016/j.jwpe.2022.102801;

https://doi.org/10.1016/j.jece.2023.109273.

Materials and methods

§  Authors should report the purity, the chemical formula and suppliers of each used chemicals in the study.

§  In “Characterization of biochar” section authors should add some details on the characterization equipment and analysis conditions.

§  In this las the atomic absorption spectroscopy (AAS) was not mentioned.

§  The physic-chemical proprieties of used wastewater (as mentioned in the manuscript title) should be added (pH, salinity, conductivity, suspended matter, DCO and DBO5 … etc)

Results and Discussions

§  In “The effect of concentration, pH, time and dose on Cr(VI) adsorption onto CBC” authors should support results discussion with references.

§  Adsorption isotherm’s part must be supported by references.

§  Why the authors in the adsorption isotherms section only choose these three models. However, many other models can give important information’s (Dubinnin-Raduskevich model, Sips …etc)?

§  The discussion of adsorption isotherm part is poor, authors should enrich it with highlighting the adsorption type and the relationship with obtained results in characterization section.

§  Adsorption kinetics part should be support by references.

§  There is no FTIR, BET, pore volume/diameter characterization, authors should add missed characterization and revise the hole manuscript.

§  An adsorption mechanism of Cr(VI)  on the biochar should be add.

Author Response

(The authors gave the same response as above.)

Reviewer 4 Report

The aim of this interesting research work is to investigate a biochar sample modified by amide incorporation to efficiently treat a chromium polluted wastewater. The effects of some important operating variables on metal removal have been also assessed through an OFAT approach. Finally, kinetics and adsorption mechanism of removal process have been explored based on known models. The paper is of significant importance in the field of wastewater treatment and contributes to the current knowledge of the issue. However, there some challenging flaws that should be addressed in the modified version of the paper as follows:

1. In title, please clearly indicate type of chemical modification used, i.e. amide modification.

2. Abstract is too long and boring. Please directly reach at the goal of your research work with especial emphasis of key findings, especially numerical results.

3. Keywords must be improved. How do you think about these, “Biochar, acid pyrolysis, amide modification, wastewater treatment, kinetics, adsorption isotherm”?

4. Please update the literature review and avoid chain citation such as 17-25. Esteemed authors should clearly explore the literature and mention the technical and scientific gaps and finally, highlight the necessity and novelty of their own work.

5. Scales in Fig. 1 are not recognizable. Please use a larger image.

6. Well, section 3.2 is a real challenge. Discussions are complete but, how do you confirm the significance of effects presented in Fig. 2 without a statistical reliability analysis? How do you confirm the reproducibility of your results? I think you have enough data to perform an ANOVA on your data points and then you can find more useful results such as interaction effects, prediction model etc. Moreover, much professionally, you can use Historical Data method to prepare and present very interesting and reliable results.

7. Fig. 3 and Table 1 show that adsorption completely fit all isotherm models! Could you please present a rational discussion on this output! Maybe some other fitting evidences are needed! I may suggest the authors to check some other models available in literature.

8. Again, similar obscurity works for kinetics results with their actually the same correlation coefficients!

9. Could you please check multi-step adsorption process?

10. Fig. 5 is too big ;)

11. In section 3.8, could you provide an economic comparison for your adsorbent?

12. Is your work perfect? If not, please suggest some comments for future works.

Good luck,

minor editing

Author Response

(The authors gave the same response as above.)

Round 2

Reviewer 1 Report

After the review of the new manuscript, I only present two recommendations to the authors:

My previous comment in the material and methods section (Section 2.5) suggests incorporating the data that the authors report (in their response to the reviewer) for the better understanding of the reader.

“The isotherm study was carried out keeping the optimum pH (2), adsorbent dose (3 g/L) and time (60 min) while the kinetics study was also performed under optimum condition (pH, dose etc.) while changing the adsorption time”

Table 2

Is there an experimental pseudo first order? Or will it be necessary to rearrange the table?

Author Response

Thanks for your comments, The Table 2 The recommended text has been added to experimental section 2.5 and the Table 2 has been rearranged.

Reviewer 2 Report

The authors have largely but not completely responded satisfactorily to my comments. Specifically:

Determination of the Cr(VI) in the material. On lines 143-145 the authors state that atomic absorption spectrometry was used to determine the quantity of Cr(VI). This is insufficient detail. Do they extract the Cr(VI) from the biochar or are they measuring the residual Cr(V)) in solution after absorption on the biochar? The description of how this was done is crucial to anyone wishing to repeat the work. A complete description of the analytical procedure to determine Cr(VI) must be included in the paper.

Inconsistencies in Figure 3. The authors state that the aim of Figure 3 is to determine the optimum conditions and this I understand. The point that I was making is that in cases where the same initial parameters are being used the outcome is different. This suggests that the removal of Cr(VI) is inconsistent. The authors need to rationalise the differences in outcome for the same starting parameters.

Section 3.2. The "-1" in "cm-1" should be superscripted.

Section 3.3. I can find no mention of a tetragonal phase of Cr2O3 in the ICSD database. Ref [43] that is cited as the source for the assignment does not include any XRD data. The usual (trigonal) form of Cr2O3 does not have a strong peak at 44.12 degrees. I find the presence of Cr2O3 surprising: this almost suggests that the biochar is inducing crystallisation, rather than adsorbing discrete ions. I find the assignment of the peak at 66 degrees to metallic Cr to be very surprising. Its presence would require a reducing agent and there is none present. I suggest the authors consider alternative assignments for these peaks.

Line 467: "k2" not "K2"

Line 504: This should be Figure 5 not Figure 4.

Further minor revision is required before the paper is acceptable for publication in Molecules.

Overall quality is acceptable but could be improved. A careful proofreading to ensure that subscripts and superscripts are applied where needed is required.

Author Response

Thank you for your valuable comments, a point by point response to your comments has been appended in the word file attached here, we do hope that this time we have successfully responded to your comments. 

Reviewer 3 Report

The authors should take into consideration our latest comment regarding the introduction in the next version of the manuscript before publication

Author Response

The introduction section has been revised accordingly 

Reviewer 4 Report

The paper has been well modified and is now suitable for publication. Good luck,

Author Response

Reviewer's comments: The paper has been well modified and is now suitable for publication. Good luck

Authors response: Thanks for your valuable suggestions to improve the quality of our manuscript